# Association between Adherence to Swedish Dietary Guidelines and Mediterranean Diet and Risk of Stroke in a Swedish Population

**DOI:** 10.3390/nu14061253

**Published:** 2022-03-16

**Authors:** Esther González-Padilla, Zhen Tao, Almudena Sánchez-Villegas, Jacqueline Álvarez-Pérez, Yan Borné, Emily Sonestedt

**Affiliations:** 1Nutritional Epidemiology, Department of Clinical Sciences Malmö, Lund University, SE 21428 Malmo, Sweden; esther.gonzalez_padilla@med.lu.se (E.G.-P.); zh1677ta-s@student.lu.se (Z.T.); yan.borne@med.lu.se (Y.B.); 2Institute for Innovation & Sustainable Development in Food Chain (ISFOOD), Universidad Pública de Navarra (UPNA), 31006 Pamplona, Spain; almudena.sanchez@unavarra.es; 3CIBEROBN (Biomedical Research Networking Centre for Physiopathology of Obesity and Nutrition), Carlos III Health Institute, 28029 Madrid, Spain; jalvarez@proyinves.ulpgc.es; 4Nutrition Research Group, Research Institute of Biomedical and Health Sciences (IUIBS), University of Las Palmas de Gran Canaria, 35016 Las Palmas de Gran Canaria, Spain

**Keywords:** stroke, Mediterranean diet, Swedish dietary guidelines score, healthy diet, dietary patterns, cohort

## Abstract

Dietary factors associated with stroke risk are still rather unknown. The aim was to examine the association between adherence to healthy dietary patterns and incidence of stroke among 25,840 individuals from the Swedish Malmö Diet and Cancer Study cohort. Dietary data were obtained using a combination of a 7-day food record, diet questionnaire, and interview. A Swedish Dietary Guidelines Score (SDGS), including five dietary components based on the current Swedish dietary guidelines, and a modified Mediterranean diet score (mMDS), composed of ten dietary components, were constructed. Over a mean follow-up period of 19.5 years, 2579 stroke cases, of which 80% were ischaemic, were identified through national registers. Weak, non-significant associations were found between the dietary indices and the risk of stroke. However, after excluding potential misreporters and individuals with unstable food habits (35% of the population), we observed significant inverse association (*p*-trend < 0.05) between SDGS and mMDS and total and ischaemic stroke (HR per point for total stroke: 0.96; 95% CI: 0.92–1.00 for SDGS and 0.95; 95% CI: 0.91–0.99 for mMDS). In conclusion, high quality diet in line with the current Swedish dietary recommendations or Mediterranean diet may reduce the risk of total and ischaemic stroke.

## 1. Introduction

Stroke is one of the most common causes of death and severe disability [1]. Ischaemic stroke is the most common subtype (around 85% of all strokes) [2], while haemorrhagic stroke, including intracerebral haemorrhage (10%) and subarachnoid haemorrhage (5%), is less common, these subtypes being associated with very high mortality rates and severe sustained disability [3,4]. In 2019, stroke remained the third-leading cause of death and disability combined (5.7% of total DALYs) globally [5]. National health systems in Western nations spend an average of 0.27% of GDP on stroke, whereas stroke contributes to around 3% of overall health-care costs [6]. Due to the high costs of stroke, prevention is paramount.

Risk factors for stroke include smoking, hypertension, diabetes, hypercholesterolemia, being overweight, physical inactivity, heavy alcohol consumption, psychosocial factors, and atrial fibrillation [7,8,9]. Since the different stroke subtypes have different pathological pathways, it is important to examine them separately [7]. Diabetes has been found to increase the risk of ischaemic stroke and intracerebral haemorrhage but reduce the risk of subarachnoid haemorrhage [7]. In addition, obesity was found to be associated with increased risk of ischaemic and reduced risk of haemorrhage stroke [7]. Diet constitutes a major modifiable risk factor; however, which dietary factors are associated with stroke risk are still rather unknown. Some studies have shown a positive association between red or processed meat consumption and risk of total or ischaemic stroke [10], and a negative association for fruit and vegetables [11], and dietary fibre [12]. 

Investigating diet in relation to diseases is a complex process. Dietary patterns instead of single nutrients or foods may capture the intricate combinations of foods consumed and synergistic effects between nutrients [13]. The Mediterranean diet, characterised by a high consumption of plant-based foods and olive oil, a moderate consumption of alcohol, and limited consumption of meat [14], has been associated with better cardiovascular health in numerous studies [15]. However, very few studies have investigated adherence to a Mediterranean diet in relation to cardiovascular disease (CVD) in Nordic populations. A diet quality index based on adherence to the Swedish dietary recommendations (which encourage intake of fruits, vegetables, fish, and fibre, while discouraging intake of sugar and saturated fat) has been previously associated with lower incidence of CVD (i.e., myocardial infarction, ischaemic stroke, or death from ischaemic heart disease) in the Malmö Diet and Cancer study (MDCS) [16]. However, there is a lack of studies examining the association between adherence to the Swedish dietary recommendations and risk of overall stroke and its subtypes. 

The aim of this study was to explore the association between two healthy dietary patterns, a healthy diet based on the current Swedish dietary guidelines and a modified Mediterranean diet, in relation to total stroke and subtypes of stroke in a prospective population-based cohort.

## 2. Materials and Methods

### 2.1. Subjects and Data Collection

Our study sample was selected from the MDCS, a population-based cohort [17]. The baseline examination of the MDCS took place between 1991 and 1996, and included a comprehensive dietary assessment, a self-administered questionnaire regarding lifestyle and socioeconomic factors, and anthropometric measurements taken onsite by trained personnel. All men born between 1923 and 1945 and all women born between 1923 and 1950 who resided in Malmö (southern Sweden) at the time of the baseline examination (*n* = 74,138) were invited to participate. Exclusion criteria were limited to insufficient knowledge of the Swedish language and mental incapacity. From all the participants meeting the inclusion criteria (*n* = 68,905), 30,446 individuals completed at least one part of the baseline examination and 28,098 individuals had complete information on dietary habits (40.8% participation rate; 38.3% for men and 42.6% for women). The MDCS was approved by the Ethical Committee at Lund University (LU 51–90) and all participants provided a written informed consent. We excluded participants with a history of stroke or coronary events (*n* = 855), diabetes at baseline (*n* = 1141), or missing data on any of the covariates (*n* = 262), resulting in a study sample of 25,840 individuals (Figure 1).

### 2.2. Dietary Data Collection and Construction of Scores

The dietary data collection for MDCS comprised a three-part modified diet history method [18,19], consisting of: (1) a 7-day food diary collecting information from hot meals (usually lunch and dinner), cold beverages, and supplement intake; (2) a 168-item food frequency questionnaire collecting information from regularly consumed products (breakfast, snacks, and other items not covered by the food diary) and hot beverages, covering the previous 12 months; (3) a 45–60 min interview with trained personnel collecting additional information on cooking methods and portion sizes, and checking that there was no overlapping information between the diary and questionnaire. Portion sizes were estimated with the help of a picture leaflet with four portion sizes references for 48 food items. Trained project staff gave during the first visit detailed instructions on the dietary data collection procedure. The average daily food intake data obtained from the combined methods were merged with a nutrient database, which was based on the Swedish Food Database (PC KOST-93) [20,21] to calculate nutrient intake. This combined method was validated against an 18-day weighted food record [19]. A relatively high relative validity was shown with energy-adjusted Pearson correlation coefficients (men/women): protein (0.54/0.53), fat (0.64/0.69), carbohydrates (0.66/0.70), fibre (0.74/0.69), sugar (0.60/0.74), vegetables (0.65/0.53), fruits (0.60/0.77), meat (0.84/0.92), fish (0.35/0.70), and wine (0.53/0.63) [18,19].

A Swedish dietary guidelines score (SDGS) was designed to portray a healthy dietary pattern according to the Swedish food-based dietary guidelines [22], which are based on the Nordic nutrition recommendations [23]. According to the Swedish dietary guidelines, consumption of vegetables, fruits, berries, fish, nuts, and seeds should be increased, while consumption of red and processed meat, salt, added sugar, and alcohol should be decreased. In addition, it is recommended to consume whole grains instead of refined grains, vegetable fats instead of butter, and low-fat dairy instead of high-fat dairy. The SDGS is composed of five dietary components: (1) fibre intake (g/MJ of non-alcoholic energy intake), (2) fish and seafood intake (g/day of fish, shellfish, fish preserves, and other fish products), (3) fruit and vegetable intake (g/day of total intake of fruits, berries, vegetables, fruit juices, and vegetable juices), (4) added sugar intake (E%, estimated by totaling the intake of monosaccharides and sucrose from the whole diet and then subtracting the intake of monosaccharides and sucrose from the main sources of naturally occurring sugars; i.e., fruits, berries, vegetables, and fruit juices), and (5) red and processed meat intake (g/day of beef, pork, lamb, game, sausages, charcuteries, and other red meat products). Fibre intake was selected mainly to reflect the recommendations to choose whole grain products. Nuts and seeds were not included because of very low intakes in MDCS, and salt was not included because of lack of reliable data. The SDGS used for this study is an updated version of a previously designed index for the MDCS population [24]. Saturated fat and polyunsaturated fat intake were not included in the SDGS as the previous index because very few of the participants of the MDCS fulfilled the recommendation of consuming below 10% of energy of saturated fat. The following recommended intake levels were used: >2.4 g/MJ for fibre, >300 g/week for fish and shellfish, >400 g/day for fruit and vegetables, <10% energy intake for added sugar, and <500 g/week for red and processed meat. Participants adhering to the recommended intake level were assigned the score of one point for each component. The overall score for the SDGS could then range from 0 (does not meet any recommendations) to 5 (meets all recommendations). Based on the SDGS scores, the participants were regrouped into low adherence (0–1 points), moderate adherence (2–3 points), or high adherence (4–5 points).

A modified Mediterranean diet score (mMDS) was created based on the original 14-item score used for the Prevención con Dieta Mediterránea (PREDIMED) study [25]. The original Mediterranean Diet Adherence Screener was validated as an accurate measurement of adherence to Mediterranean diet [26]. The use of sofrito sauce was not included in the mMDS as it is not commonly consumed in the Nordic countries. Neither were the questions regarding whether the participants used olive oil as their main culinary fat, nor whether they favoured the consumption of white meat over red meat, since we only focused on quantitative cut-off questions. Additionally, the consumption of olive oil was added to the intake of other vegetable oils because the consumption of these oils is usually low in the Nordic population. Lastly, the intakes of vegetables and legumes were used as a combined variable because they were presented as such in the MDCS dietary database. Thus, the mMDS was composed of ten food components: (1) fish and seafood (fish, shellfish, fish preserves, and other fish products), (2) fruit and berries (fruits, citrus fruits, berries, fruit juices, and citrus juices), (3) vegetables and legumes (vegetables including legumes, and vegetable juice), (4) nuts and seeds (nuts, seeds, almond paste, and other nut products), (5) vegetable oils (olive oil, rapeseed oil, corn oil, sunflower seed oil, and other vegetable oils), (6) wine, (7) butter, cream, and margarine, (8) red and processed meat (beef, pork, lamb, game, sausages, charcuteries, and other meat products), (9) soda drinks (carbonated and non-carbonated sodas, whether caloric or non-caloric), and (10) sweets and pastries (biscuits, cakes, pies, other baked goods, sweets, and chocolate). The recommendations established for these items were based on the recommendations set by Martinez-Gonzalez et al. [25] as seen in Appendix A. The intakes (g/day) were transformed into servings/week using the serving sizes estimated for the PREDIMED study (Appendix A). The participants meeting the recommendations were assigned a score of one point for each item. The overall score for the mMDS could then range from 0 (does not meet any recommendations) to 10 (meets all recommendations). Based on the mMDS scores, the participants were regrouped into low adherence (0–1 points), moderate adherence (2–4 points), or high adherence (5–10 points).

### 2.3. Endpoint Ascertainment

Participants were followed until diagnosis of event, death, emigration from Sweden, or the end of the follow up period on 31 December 2016. The mean follow-up time was 19.5 years. The endpoint was ascertained through the Hospital Discharge Register. The Stroma register was also used as a validation tool for cases that occurred before 2010 [27]. Stroke was defined according to the 9th edition of International Classification of Diseases (ICD-9) based on the following codes: ischaemic stroke (ICD-9 code 434), subarachnoid haemorrhage (ICD-9 code 430), intracerebral haemorrhage (ICD-9 code 431), and unspecified stroke (ICD-9 code 436). The Statistics in Sweden, Swedish National Tax Agency and the National Board of Health and Welfare were used to obtain data on death and emigration from Sweden. 

### 2.4. Other Variables

The age and sex of the participants were collected from the Swedish registry through their personal identification number. A questionnaire was given to the participants upon entry of the study to collect information regarding lifestyle and socioeconomic factors, as well as their past medical history. The information included some of the covariates used for this study, such as smoking habits (never smoked, former smoker, and current smoker), educational level (elementary school or less, primary and secondary school, upper secondary school, university education without or with degree), and leisure time–physical activity (five predefined groups based on Metabolic Equivalent Task (MET) hours per week, described elsewhere [28]). Alcohol consumption data was collected from the questionnaire and food diary (zero consumers and sex-specific quintiles of consumption, described elsewhere [29]). Lastly, body mass index (BMI) (kg/m^2^) was calculated from the measured height and weight of the participants during the baseline assessment [17,30]. Potential energy misreporters were identified based on their reported energy intake to calculated basal metabolic rate ratio being outside of the 95% confidence interval (CI) of the physical activity level (estimated from information on physical activity during leisure-time, at work, household work, and estimated sleeping hours) [31]. Individuals were classified as diet changers if they answered yes to the following question in the questionnaire: “Have you substantially changed your eating habits because of illness or for some other reason?” [32].

### 2.5. Statistical Analyses

Baseline characteristics across the diet scores were calculated, using analysis of variance for continuous and the Chi-square test for categorical variables. Cox proportional hazard regression was used to obtain hazard ratios (HR) and 95% confidence intervals (CI) for the two dietary scores (SDGS and mMDS) and the three outcomes (total stroke, ischaemic stroke, and haemorrhagic stroke). Years of follow up was used as the time variable and the low adherence group was used as reference. The scores were also run as continuous variables. Three models were used for adjustments. Model 1 was adjusted for age, sex, interview method (45 or 60 min interview), season when dietary data collection took place, and total energy intake (MJ/day). Model 2 was further adjusted for alcohol consumption, smoking habits, educational level, and leisure time–physical activity. Lastly, Model 3 was further adjusted for BMI and considered our main model. Additionally, we explored the risk of stroke for each dietary component of both scores individually. Sensitivity analyses were run to exclude potential energy misreporters and participants that had made substantial changes to their diets at any time before the baseline examination. The main purpose behind this was to account for the limitations of a single dietary measurement by eliminating the potentially unreliable reporters. We also examined the interactions of both diet scores with age and sex with our three outcomes using our main model. All analyses were performed using IBM SPSS Statistics (version 27; IBM Sweden AB, Stockholm, Sweden).

## 3. Results

Regarding the SDGS, we observed that 2.7% adhered to the recommendations for all five dietary components, and 13.6 % did not adhere to any of the recommendations (Figure 2). Only 21.7% of the population met the recommendation for red and processed meat (<500 g/week), 32.9% for dietary fibre (>2.4 g/MJ), and 35.5% for fruit and vegetables (>400 g/day), while 54.3% met recommendations for added sugar (<10 E%), and 45.4% for fish (>300 g/week) (Figure 3). 

Regarding the mMDS, none of the participants met all the requirements for the score. Only 7.6% had five or more points and were categorised as high adherence, while 15.0% did not adhere to any of the ten food components (Figure 2). Very few of the participants reached the cut-off for vegetable oils (four tablespoons per day; four participants), nuts and seeds (>90 g per week; 2.9%), butter, margarine, and cream (<12 g per day; 3.3%), and vegetables and legumes (>300 g per day; 11.1%) (Figure 3).

The Spearman correlation coefficient between the two dietary scores was 0.57. All the included dietary components showed a significant trend across the indices. Individuals with high adherence of SDGS or mMDS were more often females, had a university degree, had a higher level of physical activity, and were less often smokers. They were also more likely to have changed their diet in the past or to have under-reported their energy intake. While the numbers of participants consuming high amounts of alcohol were similar across the SDGS groups, there was higher frequency of high alcohol consumers among the participants scoring high on the mMDS (41.9%) when compared to the groups scoring low (10.4%). There was no major difference regarding age or BMI across the categories (Table 1 and Table 2). 

Over a mean follow up period of 19.5 years, 2579 (10%) cases of stroke were recorded, of which approximately 80% were ischaemic (2104 cases). The incidence of stroke and subtypes were lower in the group with high adherence when compared with the group with low adherence to the diets. In the basic model, both scores were statistically significant associated with total (*p*-trend for SDGS = 0.001 and mMDS = 0.008) and ischaemic stroke (*p*-trend for SDGS = 0.001 and mMDS = 0.03) (Table 3 and Table 4). However, in the multivariable models, these associations were attenuated, and we found no significant association for any type of stroke. When comparing the highest versus lowest adherence to the scores regarding risk of total stroke, we found a HR of 0.89 (95% CI = 0.78–1.02; *p*-trend = 0.10) for SDGS and 0.92 (95% CI = 0.76–1.11; *p*-trend = 0.18) for mMDS. In sensitivity analyses, where we excluded those participants who were classified as diet changers and potential energy misreporters (35% of participants), the associations were somewhat stronger and statistically significant (HR = 0.87; 95% CI = 0.72–1.04; *p*-trend = 0.04 for SDGS and HR = 0.81; 95% CI = 0.63–1.05; *p*-trend = 0.02 for mMDS) (Table 3 and Table 4).

When examining each dietary component included in the scores individually, we found weak protective associations for total and ischaemic stroke for all components in the SDGS, but a statistically significant association was found only for dietary fibre and total stroke after excluding potential misreporters (HR = 0.89; 95% CI = 0.79–1.00 for adherence when compared to non-adherence to the recommendation) (Table 5). For the dietary components in the mMDS, we observed a protective association for total stroke, and especially haemorrhagic stroke, among participants consuming on average more than 100 g of wine per day when compared to those that consumed less (Table 5). In addition, consumption of less than 200 g per day of sodas was associated with a lower risk of total and ischaemic stroke. Conversely, an increased risk of total stroke was found among participants consuming less than three servings per week of sweets and confectionaries. After excluding potential misreporters, there were statistically significant associations for total and ischaemic stroke for participants adhering to the recommendations for fruit and berries and soda intakes (Table 5). 

We also explored whether there was heterogeneity of effect between men and women and with different ages. We found a significant interaction between age and the mMDS for total stroke (*p*-interaction = 0.02), and ischaemic stroke (*p*-interaction = 0.009). When we split our participants based on their median age (57.3 years old), a protective association was found in the younger age group (HR = 0.93; 95% CI = 0.87–0.99 for total stroke and HR = 0.92; 95% CI = 0.85–0.98 for ischaemic stroke), but not in the older age group (*p* = 0.98 for total stroke and *p* = 0.59 for ischaemic stroke).

## 4. Discussion

In this large cohort study, non-significant associations were found between the dietary scores, reflecting adherence to the Swedish dietary guidelines and Mediterranean diet and the risk of stroke. Nevertheless, when restricting the sample to individuals with stable food habits and adequate energy reporting, the association between adherence to the scores and total stroke and ischaemic stroke was stronger, and statistically significant. However, no associations between adherence to the diets and haemorrhagic stroke were found.

The SDGS developed for this project has not been examined in relation to disease risk previously; however, this diet score is similar to the healthy Nordic diet consisting of a high intake of fish, apples and pears, cabbages, root vegetables, rye bread, and oatmeal. Adherence to a healthy Nordic diet was found to be inversely associated with the risk of total and ischaemic stroke, but not haemorrhagic stroke, in a Danish cohort [33]. Previous studies have shown a link between certain dietary components and a lower risk of stroke [34,35,36]. In our study, all dietary components of the SDGS contributed weakly to the protective association against stroke. A statistically significant inverse association was only observed for dietary fibre and total stroke, after excluding potential misreporters. Dietary fibre has been linked to reduced stroke risk in several cohort studies [37]. 

There are several studies suggesting a protective association between adherence to the Mediterranean diet and CVD [38,39]. Two CVD risk markers (total- and LDL-cholesterol concentration) were found to benefit from a Mediterranean diet in a Cochrane review of randomised trials [40]. However, there was still not enough evidence to demonstrate a benefit of clinical outcomes. Individual components of the Mediterranean diet, particularly fish, whole grains, fruit, and vegetables, have been found to have a positive effect against CVD in cohort studies [41]. A few observational studies have examined the association between the Mediterranean diet and risk of stroke [42]. In a Swedish cohort, high adherence to the Mediterranean diet was associated with lower risk of ischaemic stroke but not haemorrhagic stroke when compared to those with low adherence [43]. It is important to be aware that adherence to the Mediterranean diet was measured in different ways across these studies. While in our study we used predefined cut-offs from the PREDIMED, the study by Tektonidis et al. used median intakes for eight dietary components to calculate adherence [43].

While the climates of the north and the south of Europe are vastly different, which undoubtedly influences the types of food that can be farmed, SDGS and mMDS are similar in terms of their general guidelines. Both are considered plant-based and limit the amount of red and processed meat in the diet. The primary distinction between the two is that mMDS has more emphasis on vegetable oils and consumption of wine. Nevertheless, in our study, the number of people categorised as having high adherence to mMDS diet was very low, especially in sensitivity analyses, and the significance of the link found in our study could be mitigated. We found rather strong correlations between the two dietary scores and similar associations were found between both scores and risk of total and ischaemic stroke. 

Although the Mediterranean dietary pattern shares components with the SDGS, the dietary components found to reduce stroke risk in the scores were different in our study. In our study, a protective association for total stroke, especially haemorrhagic stroke, among participants consuming on average more than 100 g of wine per day (included in the mMDS) was observed. However, evidence for the relationship between alcohol and stroke is ambiguous. In a meta-analysis, when compared to no alcohol intake, low alcohol consumption (<15 g/day) reduced the risk of ischaemic stroke, but not haemorrhagic stroke, while excessive alcohol consumption was associated with an increased risk of total stroke [44].

Our finding that a consumption of less than 200 g per day of sodas (combining both sugar-sweetened and artificially sweetened soda) was associated with a lower risk of total and ischaemic stroke than consuming more than 200 g per day, coincides with previous observational studies which reported that drinking soda more than once a day was associated with an elevated risk of stroke [45]. We have previously found in the MDCS that consumption of more than 8 servings of sugar-sweetened soda per day was associated with increased risk of total stroke when compared to lower intakes [46]. On the contrary, increased risk of total stroke was shown among participants consuming less than three servings per week of sweets and pastries. We have previously found increased risk of total stroke among individuals consuming less than two servings per week [46]. In our study, statistically significant associations for total and ischaemic stroke were found with intakes of fruit and berries after excluding potential energy misreporters, which could be partly mediated through the effect on blood pressure from specific components found in fruit and berries, such as potassium and fibre [36].

Our study has various strengths. Firstly, the large study sample size enabled detailed sensitivity analyses to be performed. An almost complete follow-up rate reduced the potential selection bias due to systematic loss to follow-up. Another advantage of the MDCS was the inclusion of a detailed dietary assessment method. One of the benefits of using a 7-day food diary for the MDCS is that the possibility of recall bias is reduced, because participants record their food intake one meal at a time rather than all at once. The dietary data also have a high degree of validity [18,19]. Additionally, the MDCS includes extensive information regarding potential confounders. In the sensitivity analysis, diet changers and potential energy misreporters were excluded, thus stronger associations were indicated between high adherence to SDGS or mMDS and total stroke and ischaemic stroke. However, due to the influence of misreporting energy intake [31], collider bias could have been introduced by stratifying on energy intake. Therefore, we should be cautious about the interpretation of the results after these exclusions. 

On the other hand, dietary information was self-reported, which presents one limitation in our study. Individuals usually underestimate intake of less healthy foods more often and overestimate healthier food intake [31]. Moreover, the adoption of a Mediterranean-style diet is still relatively low in the Nordic countries, presumably due to cultural food preferences and food availability discrepancies. Another concern arising from this study is the dietary examination. When analysing dietary intakes, some degree of misclassification is unavoidable; however, because of the follow-up strategy, any misclassification was most likely to be nondifferential, which could cause underestimating of the underlying relationship. Furthermore, this study only included middle-aged to elderly participants from Malmö city in southern Sweden, which hampers the generalisability of our findings. Also, these findings may be influenced by several lifestyle factors, which play a significant role in the development of stroke. Even though we controlled for multiple variables in our study, we still cannot rule out the chance of potential residual confounding affecting the observed association. 

## 5. Conclusions

In conclusion, high adherence to a healthy diet based on the current Swedish dietary guidelines or a Mediterranean diet could be linked to lower risk of stroke, particularly for those participants with a more reliable dietary reporting in this large Swedish cohort study. Given the incomplete understanding of the relationships between food and stroke incidence, further research in addition to the findings of this study will contribute significantly to the future establishment of dietary guidelines for stroke prevention.

## Figures and Tables

**Figure 1 nutrients-14-01253-f001:**
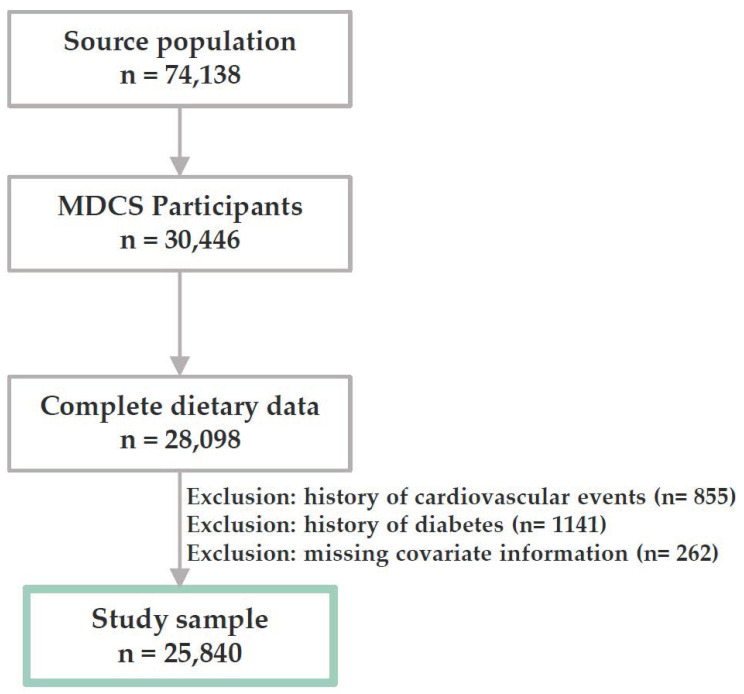
Study sample selection. MDCS: Malmö Diet and Cancer Study.

**Figure 2 nutrients-14-01253-f002:**
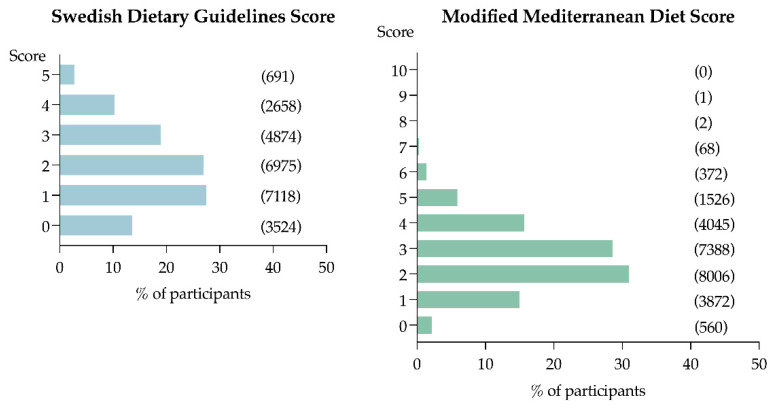
Percentage of participants according to score punctuation. Number of participants classified on their score punctuation presented in parentheses.

**Figure 3 nutrients-14-01253-f003:**
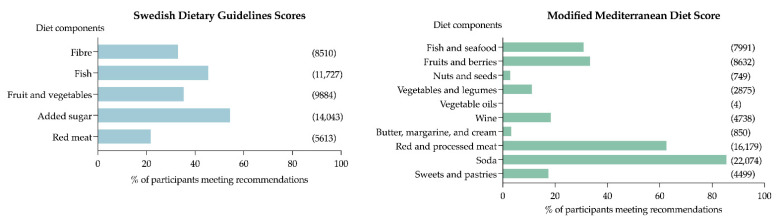
Percentage of participants meeting the recommendations for each diet component. Number of participants meeting recommendations presented in parentheses.

**Table 1 nutrients-14-01253-t001:** Baseline characteristics across Swedish dietary guidelines score (SDGS) adherence levels.

	Low (0–1 Points)	Medium (2–4 Points)	High (5–10 Points)	*p*-Value
*n*	10,642	11,849	3349	
**Mean (SD)**				
Age, years	57.5 (7.6)	58.1 (7.6)	57.9 (7.6)	<0.001
BMI, kg/m^2^	25.5 (3.9)	25.7 (3.9)	25.4 (3.9)	<0.001
Fibre, g/MJ	1.83 (0.38)	2.32 (0.59)	3.00 (0.61)	<0.001
Fish and shellfish, g/week	215 (179)	368 (253)	472 (259)	<0.001
Fruit and vegetables, g/day	271 (107)	412 (173)	584 (196)	<0.001
Added sugar, E%	12.0 (4.4)	9.19 (3.78)	7.39 (2.91)	<0.001
Red and processed meat, g/week	930 (399)	798 (431)	530 (335)	<0.001
Total energy, MJ/day	9.84 (2.70)	9.55 (2.80)	8.65 (2.33)	<0.001
Carbohydrates, E%	44.6 (5.6)	45.0 (6.4)	47.5 (5.7)	<0.001
Protein, E%	15.0 (2.3)	16.1 (2.5)	17.0 (2.7)	<0.001
Fat, E%	40.4 (5.6)	38.9 (6.3)	35.5 (5.6)	<0.001
Vitamin C, mg/MJ	9.1 (5.3)	12.1 (6.3)	17.5 (7.6)	<0.001
Vitamin D, ug/MJ	0.74 (0.23)	0.81 (0.30)	0.87 (0.33)	<0.001
Folate, ug/MJ	22.7 (5.5)	27.1 (6.7)	33.7 (8.0)	<0.001
Iron, mg/MJ	1.59 (0.32)	1.64 (0.32)	1.66 (0.30)	<0.001
Calcium, mg/MJ	114 (34)	122 (35)	134 (36)	<0.001
Potassium, mg/MJ	326 (59)	373 (69)	440 (80)	<0.001
Magnesium, mg/MJ	34.0 (4.9)	37.6 (5.7)	42.8 (6.5)	<0.001
Selenium, ug/MJ	3.51 (0.96)	4.20 (1.25)	4.87 (1.47)	<0.001
Zinc, mg/MJ	1.15 (0.19)	1.19 (0.20)	1.21 (0.20)	<0.001
**N (%)**				
Females	5899 (55.4%)	7480 (63.1%)	2692 (80.4%)	<0.001
Smokers	3743 (35.2%)	2956 (24.9%)	639 (19.1%)	<0.001
University degree	1179 (11.1%)	1889 (15.9%)	717 (21.4%)	<0.001
Highest quintile of alcohol intake	1904 (17.9%)	2426 (20.5%)	616 (18.4%)	<0.001
Low leisure-time physical activity	1287 (12.1%)	947 (8.0%)	178 (5.3%)	<0.001
Underreporting	1366 (12.8%)	1860 (15.7%)	704 (21.0%)	<0.001
Past diet change	1652 (15.5%)	2677 (22.6%)	1230 (36.8%)	<0.001

**Table 2 nutrients-14-01253-t002:** Baseline characteristics across modified Mediterranean diet score (mMDS) adherence levels.

	Low (0–1 Points)	Medium (2–4 Points)	High (5–10 Points)	*p*-Value
*n*	4432	19,439	1969	
**Mean (SD)**				
Age, years	57.3 (7.3)	58.1 (7.7)	56.6 (7.2)	<0.001
BMI, kg/m^2^	25.9 (4.0)	25.5 (3.9)	25.3 (3.7)	<0.001
Fish and shellfish, servings/week	1.50 (1.13)	2.51 (1.88)	3.94 (2.34)	<0.001
Fruit and berries, servings/week	11.9 (7.3)	18.2 (11.5)	27.8 (13.6)	<0.001
Nuts and seeds, servings/week	0.29 (0.61)	0.40 (1.04)	1.00 (2.30)	<0.001
Vegetables and legumes, servings/week	3.42 (1.56)	4.21 (2.22)	7.03 (3.41)	<0.001
Vegetable oils, servings/week	0.42 (1.00)	0.53 (1.16)	1.08 (2.35)	<0.001
Wine, servings/week	1.49 (2.30)	3.38 (4.81)	7.12 (6.69)	<0.001
Butter, margarine and cream, servings/week	38.5 (19.3)	31.9 (17.7)	24.9 (17.3)	<0.001
Red and processed meat, servings/week	9.15 (3.21)	6.15 (3.20)	4.53 (2.72)	<0.001
Soda, servings/week	6.66 (7.97)	2.43 (4.62)	1.22 (2.47)	<0.001
Sweets and pastries, servings/week	10.6 (7.1)	8.87 (6.74)	5.26 (5.78)	<0.001
Total energy, MJ/day	10.6 (2.8)	9.34 (2.67)	9.23 (2.66)	<0.001
Carbohydrates, E%	44.4 (5.5)	45.2 (6.0)	46.7 (6.9)	<0.001
Protein, E%	14.9 (2.2)	15.8 (2.5)	17.2 (3.0)	<0.001
Fat, E%	40.7 (5.6)	39.0 (6.0)	36.2 (6.8)	<0.001
Vitamin C, mg/MJ	8.4 (4.5)	11.7 (6.4)	18.0 (8.6)	<0.001
Vitamin D, ug/MJ	0.74 (0.21)	0.79 (0.28)	0.84 (0.36)	<0.001
Folate, ug/MJ	22.1 (5.3)	26.3 (6.8)	33.8 (9.6)	<0.001
Iron, mg/MJ	1.63 (0.33)	1.61 (0.31)	1.66 (0.31)	<0.001
Calcium, mg/MJ	108 (32)	122 (36)	131 (38)	<0.001
Potassium, mg/MJ	319 (58)	365 (72)	438 (93)	<0.001
Magnesium, mg/MJ	33.6 (4.8)	37.0 (5.8)	42.3 (7.6)	<0.001
Selenium, ug/MJ	3.42 (0.89)	4.04 (1.22)	4.99 (1.65)	<0.001
Zinc, mg/MJ	1.19 (0.19)	1.18 (0.20)	1.20 (0.21)	<0.001
**N (%)**				
Females	1908 (43.1%)	12,762 (65.7%)	1401 (71.2%)	<0.001
Smokers	1504 (33.9%)	5358 (27.6%)	476 (24.2%)	<0.001
University degree	354 (8.0%)	2884 (14.8%)	544 (27.6%)	<0.001
Highest quintile of alcohol intake	463 (10.4%)	3658 (18.8%)	825 (41.9%)	<0.001
Low leisure-time physical activity	523 (11.8%)	1769 (9.1%)	120 (6.1%)	<0.001
Underreporting	520 (11.7%)	3027 (15.6%)	383 (19.5%)	<0.001
Past diet change	730 (16.5%)	4146 (21.3%)	683 (34.8%)	<0.001

**Table 3 nutrients-14-01253-t003:** Association (HR and 95% CI) between Swedish dietary guidelines score and risk of stroke.

		Adherence to Swedish Dietary Guidelines Score				
		Low (0–1 Points)	Medium (2–3 Points)	High (4–5 Points)	Per Point	*p*-Trend
*n*		10,642	11,849	3349		
Years of follow-up		203,453	232,484	67,344		
Total stroke	Cases/cases per 1000 PY	1101/5.41	1189/5.11	289/4.29		
	Model 1	1.00	0.91 (0.84–0.99)	0.81 (0.71–0.93)	0.95 (0.92–0.98)	0.001
	Model 2	1.00	0.97 (0.89–1.05)	0.89 (0.78–1.02)	0.98 (0.95–1.01)	0.13
	Model 3	1.00	0.96 (0.88–1.04)	0.89 (0.78–1.02)	0.97 (0.94–1.01)	0.10
	Model 3 (excl. misreporters)	1.00	0.94 (0.85–1.04)	0.87 (0.72–1.04)	0.96 (0.92–1.00)	0.04
Ischaemic stroke	Cases/cases per 1000 PY	910/4.47	957/4.12	237/3.52		
	Model 1	1.00	0.89 (0.81–0.98)	0.81 (0.70–0.94)	0.94 (0.91–0.98)	0.001
	Model 2	1.00	0.95 (0.86–1.04)	0.90 (0.78–1.05)	0.97 (0.94–1.01)	0.14
	Model 3	1.00	0.94 (0.86–1.03)	0.90 (0.77–1.04)	0.97 (0.94–1.01)	0.10
	Model 3 (excl. misreporters)	1.00	0.91 (0.81–1.02)	0.87 (0.72–1.07)	0.95 (0.91–1.00)	0.03
Haemorrhagic stroke	Cases/cases per 1000 PY	172/0.85	209/0.90	51/0.78		
	Model 1	1.00	1.02 (0.83–1.25)	0.89 (0.64–1.22)	0.97 (0.90–1.05)	0.48
	Model 2	1.00	1.07 (0.87–1.32)	0.95 (0.69–1.31)	0.99 (0.92–1.07)	0.88
	Model 3	1.00	1.07 (0.87–1.32)	0.95 (0.69–1.31)	0.99 (0.92–1.07)	0.89
	Model 3 (excl. misreporters)	1.00	1.15 (0.90–1.48)	0.96 (0.62–1.48)	1.01 (0.91–1.11)	0.87

Model 1: adjusted for age, sex, method, season, energy intake. Model 2: adjusted for age, sex, method, season, energy intake, education, smoking, leisure time–physical activity, alcohol habits. Model 3: adjusted for age, sex, method, season, energy intake, education, smoking, leisure time–physical activity, alcohol habits, BMI. Model 3: excluding misreporters (i.e., non-adequate reporters of energy) and those who indicated a substantial change in dietary habits in the past (35% of the study sample).

**Table 4 nutrients-14-01253-t004:** Association (HR and 95% CI) between modified Mediterranean diet score and risk of stroke.

		Adherence to Modified Mediterranean Diet Score				
		Low (0–1 Points)	Medium (2–4 Points)	High (5–10 Points)	Per Point	*p*-Trend
*n*		4432	19,439	1696		
Years of follow-up		84,834	378,907	39,542		
Total stroke	Cases/cases per 1000 PY	480/5.66	1936/5.11	163/4.12		
	Model 1	1.00	0.90 (0.81–0.99)	0.84 (0.70–1.00)	0.96 (0.93–0.99)	0.008
	Model 2	1.00	0.94 (0.84–1.04)	0.92 (0.76–1.11)	0.98 (0.94–1.01)	0.18
	Model 3	1.00	0.94 (0.85–1.04)	0.92 (0.76–1.11)	0.98 (0.94–1.01)	0.18
	Model 3 (excl. misreporters)	1.00	0.89 (0.78–1.00)	0.81 (0.63–1.05)	0.95 (0.91–0.99)	0.02
Ischaemic stroke	Cases/cases per 1000 PY	396/4.67	1576/4.16	132/3.34		
	Model 1	1.00	0.89 (0.79–1.00)	0.83 (0.68–1.02)	0.96 (0.93–1.00)	0.03
	Model 2	1.00	0.93 (0.83–1.05)	0.92 (0.75–1.13)	0.98 (0.95–1.02)	0.37
	Model 3	1.00	0.94 (0.84–1.05)	0.93 (0.75–1.14)	0.98 (0.95–1.02)	0.38
	Model 3 (excl. misreporters)	1.00	0.87 (0.76–1.00)	0.79 (0.59–1.04)	0.95 (0.90–1.00)	0.03
Haemorrhagic stroke	Cases/cases per 1000 PY	74/0.87	329/0.87	29/0.73		
	Model 1	1.00	0.96 (0.74–1.25)	0.90 (0.59–1.40)	0.94 (0.87–1.02)	0.12
	Model 2	1.00	1.00 (0.77–1.30)	0.97 (0.62–1.52)	0.95 (0.87–1.03)	0.24
	Model 3	1.00	1.00 (0.77–1.30)	0.97 (0.62–1.52)	0.95 (0.87–1.03)	0.24
	Model 3 (excl. misreporters)	1.00	1.02 (0.74–1.40)	1.07 (0.60–1.94)	0.96 (0.86–1.07)	0.47

Model 1: adjusted for age, sex, method, season, energy intake. Model 2: adjusted for age, sex, method, season, energy intake, education, smoking, leisure time–physical activity, alcohol habits. Model 3: adjusted for age, sex, method, season, energy intake, education, smoking, leisure time–physical activity, alcohol habits, BMI. Model 3: excluding misreporters (i.e., non-adequate reporters of energy) and those who indicated a substantial change in dietary habits in the past (35% of study participants).

**Table 5 nutrients-14-01253-t005:** Association (HR and 95% CI) between individual dietary components in the dietary scores and risk of stroke ^1^.

Diet Score	Diet Component	Recommendation	N (%) Reaching Recommendation	HR (95% CI) for Those Reaching Recommendations					
				Whole Sample			Excluding Misreporters and Diet Changers ^2^		
				Total Stroke	Ischaemic Stroke	Haemorrhagic Stroke	Total Stroke	Ischaemic Stroke	Haemorrhagic Stroke
SDGS									
	**Encouraged**								
	Fibre	>2.4 g/MJ	8510 (32.9%)	0.92 (0.85–1.01)	0.92 (0.84–1.02)	0.94 (0.77–1.17)	0.89 (0.79–1.00)	0.90 (0.79–1.02)	0.88 (0.66–1.16)
	Fish	>300 g/week	11,727 (45.4%)	0.98 (0.90–1.06)	1.00 (0.91–1.09)	0.96 (0.79–1.17)	0.97 (0.88–1.07)	0.97 (0.87–1.09)	1.04 (0.82–1.32)
	Fruit and vegetables	>400 g/day	9884 (35.3%)	0.96 (0.88–1.05)	0.93 (0.85–1.02)	1.12 (0.91–1.37)	0.92 (0.83–1.03)	0.90 (0.80–1.01)	1.07 (0.83–1.38)
	**Discouraged**								
	Added sugar	<10%E	14,043 (54.3%)	0.99 (0.92–1.08)	0.97 (0.89–1.06)	1.08 (0.89–1.31)	0.97 (0.88–1.07)	0.94 (0.85–1.05)	1.12 (0.88–1.42)
	Red and processed meat	<500 g/week	5613 (21.7%)	0.94 (0.85–1.04)	0.97 (0.86–1.08)	0.81 (0.63–1.04)	0.94 (0.82–1.08)	0.95 (0.81–1.10)	0.90 (0.65–1.25)
mMDS									
	**Encouraged**								
	Fish and seafood	≥3 svg/week	7991 (30.9%)	1.02 (0.94–1.11)	1.04 (0.95–1.15)	0.97 (0.78–1.19)	1.00 (0.90–1.11)	1.00 (0.89–1.12)	1.04 (0.81–1.33)
	Fruits and berries	≥21 svg/week	8632 (33.4%)	0.94 (0.86–1.03)	0.94 (0.86–1.04)	0.88 (0.71–1.08)	0.89 (0.80–1.00)	0.89 (0.78–1.00)	0.87 (0.67–1.13)
	Nuts and seeds	≥3 svg/week	749 (2.9%)	0.78 (0.60–1.01)	0.77 (0.57–1.04)	0.81 (0.43–1.52)	0.79 (0.57–1.10)	0.81 (0.57–1.16)	0.77 (0.34–1.73)
	Vegetables and legumes	≥7 svg/week	2875 (11.1%)	0.96 (0.83–1.09)	0.95 (0.82–1.11)	0.97 (0.70–1.33)	0.93 (0.77–1.13)	0.91 (0.74–1.12)	1.05 (0.69–1.61)
	Vegetable oils	≥28 svg/week	4 (0%)	-	-	-	-	-	-
	Wine	≥7 svg/week	4738 (18.3%)	0.87 (0.75–1.00)	0.92 (0.79–1.07)	0.65 (0.45–0.93)	0.88 (0.75–1.04)	0.93 (0.78–1.12)	0.66 (0.43–1.01)
	**Discouraged**								
	Butter, margarine, and cream	<7 svg/week	850 (3.3%)	1.14 (0.91–1.42)	1.15 (0.90–1.48)	1.14 (0.66–1.96)	1.17 (0.81–1.69)	1.09 (0.71–1.66)	1.72 (0.81–3.67)
	Red and processed meat	<7 svg/week	16,179 (62.6%)	0.97 (0.88–1.06)	0.97 (0.88–1.07)	1.06 (0.85–1.33)	0.95 (0.85–1.06)	0.94 (0.83–1.06)	1.07 (0.81–1.41)
	Soda	<7 svg/week	22,074 (85.4%)	0.89 (0.80–0.99)	0.89 (0.79–1.00)	0.86 (0.66–1.12)	0.83 (0.73–0.94)	0.81 (0.70–0.93)	0.88 (0.64–1.22)
	Sweets and pastries	<3 svg/week	4499 (17.4%)	1.12 (1.01–1.24)	1.12 (1.00–1.26)	1.10 (0.85–1.42)	1.13 (0.98–1.30)	1.15 (0.98–1.34)	1.01 (0.71–1.45)

^1^ Adjusted for age, sex, method, season, energy intake, education, smoking, leisure time–physical activity, alcohol habits, BMI. ^2^ Excluding non-adequate reporters of energy (i.e., potential misreporters) and those who indicated a substantial change in dietary habits in the past.

## Data Availability

The dataset presented in this article are not readily available because of ethical and legal restrictions. Requests to access the dataset should be directed to the Chair of the Steering Committee for the Malmö cohorts, see instructions at https://www.malmo-kohorter.lu.se/malmo-cohorts (last accessed on 28 February 2022).

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
