# Peer review of "Association between Adherence to Swedish Dietary Guidelines and Mediterranean Diet and Risk of Stroke in a Swedish Population"

_nutrients, 2022, doi:10.3390/nu14061253_

Round 1
Reviewer 1 Report
This article provided important and relevant information regarding, the associations between consumption of a healthy diet and the incidence of stroke. The inclusion/comparison of both a Swedish diet and a Mediterranean diet was interesting especially, as they are both considered to be healthy and disease preventative. The use of a shorter (meal-by-meal) dietary record was important, as it is easy to forget food intake amounts or items if the food diary is long.
It is of specific relevance in today's population, as the incidence of cardiovascular disease, stroke and obesity is increasing throughout the world and identification of dietary guidelines/habits that reduce risk of disease can be useful.
This was an interesting article and quite relevant to current health issues.
Perhaps include information relating to how stable food reporting and adequate energy reporting was determined.
Was there a trial period to instruct participants on the correct reporting of food intake.
Were there significant associations with obesity and stroke, smoking and stroke, consumption of alcohol and stroke in the current study?
Author Response
This article provided important and relevant information regarding, the associations between consumption of a healthy diet and the incidence of stroke. The inclusion/comparison of both a Swedish diet and a Mediterranean diet was interesting especially, as they are both considered to be healthy and disease preventative. The use of a shorter (meal-by-meal) dietary record was important, as it is easy to forget food intake amounts or items if the food diary is long.
It is of specific relevance in today's population, as the incidence of cardiovascular disease, stroke and obesity is increasing throughout the world and identification of dietary guidelines/habits that reduce risk of disease can be useful.
This was an interesting article and quite relevant to current health issues.
Perhaps include information relating to how stable food reporting and adequate energy reporting was determined.
ANSWER: We have clarified the way that stable food reporting and adequate energy reporting was determined (line 196-203).
Was there a trial period to instruct participants on the correct reporting of food intake.
ANSWER: We have added information regarding the instructions to participants (line 106-107).
Were there significant associations with obesity and stroke, smoking and stroke, consumption of alcohol and stroke in the current study?
ANSWER: Obesity and smoking was associated with increased risk of stroke. For alcohol, we found a U-shaped association.
Reviewer 2 Report
Dear Authors,
Thank you very much for a well-designed study and interesting manuscript. The topic is important and adds to the highest quality research literature in nutrition epidemiology and public health. I have only a few minor comments for your consideration.
Please add a paragraph explaining the main features of Swedish dietary guidelines and Nordic nutrition recommendations to the Introduction section.
Please provide a response rate in the baseline data collection (Section 2.1.) in men and women populations. Also, I think the information provided in the supplementary materials Figure S1 is important and should be moved to the main text.
The description of the dietary assessment is very well presented.
Please provide a mean time of the follow-up in section 2.3. Also, it would be helpful to explain here the sources and information collection on lost to follow-up (migration, etc.).
Next, the reader might wonder if characteristics presented in Tables 1 and 2 across low, medium and high adherence groups demonstrate significant differences? (the chi-square test, ANOVA).
Author Response
Dear Authors,
Thank you very much for a well-designed study and interesting manuscript. The topic is important and adds to the highest quality research literature in nutrition epidemiology and public health. I have only a few minor comments for your consideration.
Please add a paragraph explaining the main features of Swedish dietary guidelines and Nordic nutrition recommendations to the Introduction section.
ANSWER: We have added a sentence in the introduction explaining the Swedish dietary guidelines (line 63-64).
Please provide a response rate in the baseline data collection (Section 2.1.) in men and women populations. Also, I think the information provided in the supplementary materials Figure S1 is important and should be moved to the main text.
ANSWER: We have added information on response rate for men and women (line 85). We have also included Figure S1 in the main text (new Figure 1).
The description of the dietary assessment is very well presented.
Please provide a mean time of the follow-up in section 2.3. Also, it would be helpful to explain here the sources and information collection on lost to follow-up (migration, etc.).
ANSWER: We have added mean time of follow-up in section 2.3 (line 174-175). We have also added information on death and emigration from Sweden (line 180-182). The registers that were used include all residents of Sweden, and there was therefore no loss to follow-up during registry linkage. 219 of the participants had emigrated from Sweden and were therefore censored at the date of emigration.
Next, the reader might wonder if characteristics presented in Tables 1 and 2 across low, medium and high adherence groups demonstrate significant differences? (the chi-square test, ANOVA).
ANSWER: We have added p-value in Table 1 and Table 2, and added the methods used in the statistical analyses section (line 204-205).